



# Aerodynamic characterisation of a thrust-scaled IEA 15 MW wind turbine model: Experimental insights using PIV data

Erik Fritz[1,2], André Ribeiro[2], Koen Boorsma[1], and Carlos Ferreira[2]

[1]Wind Energy, TNO Energy Transition, Petten, Netherlands
[2]Faculty of Aerospace Engineering, Technical University of Delft, Delft, Netherlands

**Correspondence:** Erik Fritz (e.fritz@tno.nl)

**Abstract.** This study presents results from a wind tunnel experiment on a three-bladed horizontal axis wind turbine. The model turbine is a scaled-down version of the IEA 15 MW reference wind turbine, preserving the non-dimensional thrust distribution along the blade.

Flow fields were captured around the blade at multiple radial locations using Particle Image Velocimetry. In addition to these flow fields, this comprehensive dataset contains spanwise distributions of bound circulation, inflow conditions and blade forces derived from the velocity field. As such, the three blades' aerodynamics are fully characterised. It is demonstrated that the lift coefficient measured along the span agrees well with the lift polar of the airfoil used in the blade design, thereby validating the experimental approach.

This research provides a valuable public experimental dataset for validating low to high-fidelity numerical models simulating state-of-the-art wind turbines. Furthermore, this article establishes the aerodynamic properties of the newly developed model wind turbine, creating a baseline for future wind tunnel experiments using this model.

## Nomenclature

| **Latin letters** | continues on next page... |
| --- | --- |
| $a, a'$ | Axial and tangential induction factor |
| $C_c$ | Chord scaling constant |
| $C_T$ | Thrust coefficient |
| $c$ | Chord |
| $c_l, c_d$ | Lift and drag coefficient |
| $c_l^0$ | Lift coefficient at zero angle of attack |
| $D$ | Rotor diameter, drag force |
| $D_{root}$ | Diameter of the blade root section |
| $\boldsymbol{F}$ | Force vector |
| $F_N, F_T$ | Normal and tangential force |





| **Latin letters** | ...continued |
|---|---|
| $\boldsymbol{I}$ | Identity matrix |
| $Kl$ | Lift slope |
| $L$ | Lift force |
| $\mathscr{N}$ | Dimensional constant |
| $\boldsymbol{n}$ | Normal vector |
| $R$ | Blade tip radius |
| $\text{Re}_c$ | Chord Reynolds number |
| $r$ | Radial coordinate |
| $r_{root}$ | Blade root radius |
| $S, S_B$ | Outer and inner boundary curve of a control volume |
| $t$ | Time |
| $U_\infty$ | Free stream velocity |
| $\boldsymbol{u}$ | Velocity vector |
| $u, v$ | Velocity components |
| $V_{rel}$ | Relative inflow velocity |
| $V_{rot}$ | Rotational velocity |
| $\boldsymbol{x}$ | Position vector |

| **Greek letters and other symbols** | |
|---|---|
| $\alpha$ | Angle of attack |
| $\beta$ | Blade pitch angle |
| $\Gamma$ | Circulation |
| $\gamma$ | Flux term |
| $\lambda$ | Tip speed ratio |
| $\lambda_L$ | Geometric scaling factor |
| $\rho$ | Density of air |
| $\boldsymbol{\tau}$ | Reynolds stress tensor |
| $\phi$ | Inflow angle |
| $\omega$ | Angular velocity |
| $\boldsymbol{\omega}$ | Vorticity vector |
| $\nabla$ | Nabla operator |



| Subscripts | |
|---|---|
| $CV$ | Control volume |
| $ind$ | Induced |
| $KJ$ | Kutta-Joukowski |
| $M$ | Model |
| $O$ | Original |

## 1   Introduction

Wind tunnel experiments are vital in progressing horizontal axis wind turbine (HAWT) technology. They help in improving the understanding of, e.g. the turbine's aerodynamic, aeroelastic or acoustic characteristics. Equally important, the gathered data can be used to validate and improve numerical models that aim to simulate reality as closely as possible.

In light of these two goals, arguably, the two most relevant experiments on HAWTs are the Unsteady Aerodynamics Experiment (UAE) and the Model Rotor Experiment in Controlled Conditions (MEXICO). NREL executed the UAE in multiple phases. While Phases I - IV, conducted between 1989 and 1997, were field experiments (Butterfield et al., 1992; Simms et al., 1999), Phase VI was a wind tunnel experiment conducted in 2000. A two-bladed rotor of 10 m diameter was heavily instrumented and placed in the NASA Ames wind tunnel (Hand et al., 2001). The MEXICO experiment was conducted in 2006 in the German-Dutch Wind Tunnel (DNW). Detailed aerodynamic measurements, including pressure, loads and 3D flow field characteristics using Particle Image Velocimetry (PIV) were taken on a three-bladed rotor with 4.5 m diameter (Schepers and Snel, 2007; Boorsma and Schepers, 2009). Its successor project "New Mexico" was conducted in 2014 to obtain additional data (Boorsma and Schepers, 2015). The results of these two experimental campaigns have been analysed in great detail and have been used for the validation/calibration of simulation tools of varying fidelity. For an extensive review of the literature related to these two experiments, the reader is referred to the work of Schepers and Schreck (2018).

Given the success of these two experiments, the existing databases were extended by conducting further experiments on scaled versions of the two rotors. The wake of a 1:8 scaled version of the UAE Phase VI rotor was measured using PIV by Xiao et al. (2011). At the Korean Aerospace Research Institute (KARI), Cho and Kim (2014) tested the Reynolds number effect on torque and power on a 1:5 scaled model of the UAE Phase VI turbine. Comparable experiments were done by the same researchers for a 2:4.5 scaled version of the MEXICO rotor (Cho and Kim, 2012). The Spanish National Institute for Aerospace Technology (INTA) tested a 1:4 scaled MEXICO rotor. Results of the scaled models are compared against the original MEXICO data in IEA Task 29 by Schepers et al. (2012).

Complementary to experimental investigations, HAWTs are studied extensively using numerical simulations. To enable numerical benchmarks between different simulation tools and to facilitate collaboration between academic and industrial research, multiple reference wind turbine (RWT) models have been developed in recent years, e.g. the NREL 5 MW RWT (Jonkman et al., 2009), the DTU 10 MW RWT (Bak, 2013) and the IEA 15 MW RWT (Gaertner et al., 2020). While not representing existing wind turbines, these open-source reference models reflect current trends and developments of HAWT technology.





Wind tunnel campaigns with scaled versions of these reference wind turbines have been conducted to provide experimental datasets that can be used to validate numerical simulations. Fontanella et al. (2021a) ran experiments on a scaled DTU 10 MW wind turbine mimicking the motions of a floating offshore wind turbine (FOWT). In addition to load cell measurements on the turbine, the wake was characterised using PIV measurements. Similar experiments were conducted by Taruffi et al. (2024), extending the mimicked floater motions to six degrees of freedom and larger amplitudes and frequencies. Fontanella et al. (2022)

performed another set of experiments on a 1:100 scaled model of the IEA 15 MW RWT developed by Allen et al. (2020). Here, rotor loads were measured using load cells, and the wake was characterised using hot wire velocity measurements. A 1:70 scaled model of the IEA 15 MW was tested by Kimball et al. (2022) with a focus on verifying thrust and torque curves and validating the utilised pitch controller. While these studies on scaled-down versions of the RWTs provide valuable data regarding rotor-level aerodynamics, they lack more detailed data on the blade-level.

Such blade-level data can be obtained using non-intrusive measurement techniques such as Laser Doppler Velocimetry (LDV) or Stereoscopic Particle Image Velocimetry (SPIV). Phengpom et al. (2015a, b, 2016) studied the flow field in the direct vicinity of the blade using LDV. Akay et al. (2013) researched the vortex structure around the blade root of a two-bladed wind turbine with a 2 m diameter based on SPIV measurements. Similarly, Lignarolo et al. (2014) investigated the wake development of a smaller model (two blades, 0.6 m diameter) focusing on the tip vortices. Continuing this line of research, the generation

of the tip vortex was investigated in more detail on a different two-bladed wind turbine model of 2 m diameter by Micallef et al. (2014, 2015). Furthermore, and most relevant to the present work, SPIV was employed to derive the spanwise blade load distribution of a HAWT in axial and yawed inflow by del Campo et al. (2014, 2015).

The present work studies the spanwise aerodynamic characteristics of a 1:133 scaled model of the IEA 15 MW RWT, thus, of the most recent available reference wind turbine. SPIV is used to measure the flow field around various radial sections of

the blade and, consequently, to derive the spanwise aerodynamic properties of this model wind turbine. By characterising the blades in terms of induction values, inflow angle and angle of attack, circulation, and blade loads, this study provides a more complete dataset of blade-level aerodynamics than previous wind tunnel experiments. As such, this research aims to enable further multi-fidelity numerical benchmarking as well as to establish a reference dataset that can be used as a starting point for future experimental studies on this wind tunnel model turbine.

This paper is built up as follows: Section 2 describes the scaling approach used to develop the wind tunnel model's blades. Furthermore, details of the experimental setup are given and the equations used to derive blade aerodynamic quantities from the measured flow fields are provided. Section 3 initially details the challenge of estimating deviations from the design twist distribution. It then presents the results of the experimental campaign in terms of flow fields, distributed blade aerodynamics and airfoil polars. Finally, conclusions are drawn in Section 4 and an outlook at further research is given.





## 2 Methodology

### 2.1 Scaled wind turbine model

The model HAWT tested in this experiment is a scaled version of the IEA 15 MW RWT (Gaertner et al., 2020), preserving non-dimensional thrust. The main model characteristics are given in Table 2 alongside their full-scale equivalents.

| Parameter | | IEA 15 MW RWT | | Wind tunnel model | |
|---|---|---|---|---|---|
| Rotor diameter | $D$ | 240 | m | 1.8 | m |
| Blade root radius | $r_{root}$ | 3 | m | 0.06 | m |
| Design tip speed ratio | $\lambda$ | 9 | - | 9 | - |

**Table 2.** Specifications of the IEA 15 MW RWT and the scaled wind tunnel model

Multiple challenges occur when creating a scaled-down wind tunnel model of a wind turbine. Arguably, the largest challenge lies in the fact that the chord Reynolds number $Re_c$ present on a full-scale wind turbine can generally not be achieved in a wind tunnel. A difference in $Re_c$ of multiple orders of magnitude necessitates the use of airfoils designed explicitly for low Reynolds numbers. One such airfoil is the *SD7032* airfoil, which has a maximum relative thickness of 10 % and was characterised experimentally by Fontanella et al. (2021b). The lift and drag coefficient of the *SD7032* airfoil for different Reynolds numbers is given in Figure 1. A characteristic of this airfoil making it useful for small-scale wind turbines is the relative insensitivity of the lift polar to the Reynolds number over a large range of angles of attack. The well documented wind tunnel polars as well as the airfoil's application in comparable wind tunnel campaigns (Kimball et al., 2022; Fontanella et al., 2022) motivated the choice for the *SD7032* airfoil.

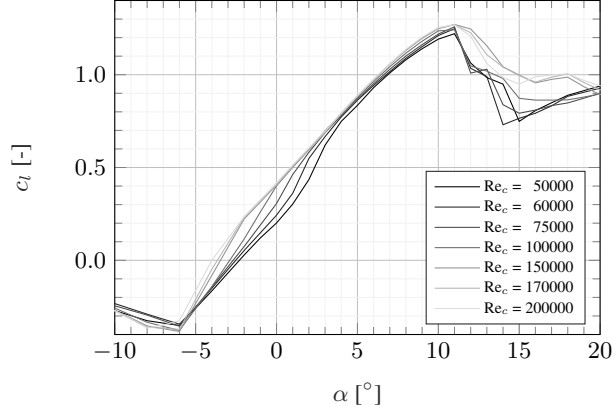
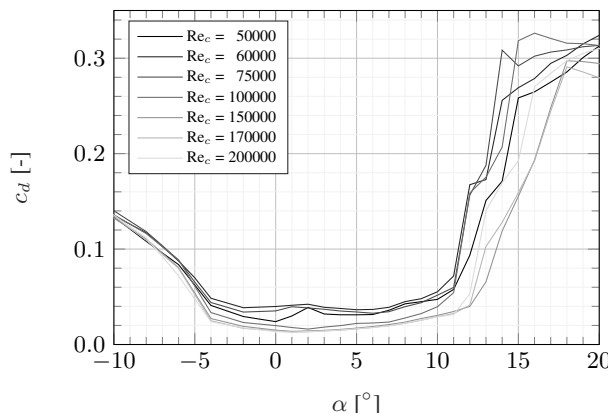

**Figure 1.** Lift coefficient $c_l$ and drag coefficient $c_d$ of the *SD7032* airfoil for varying Reynolds numbers (Fontanella et al., 2021b)





Instead of using various airfoils along the span, the developed model blades are defined by this single airfoil, which transitions into a cylindrical section at the blade root. The polars of the wind tunnel model differ from those of the airfoils used
on the full-scale turbine. Thus, even at identical angles of attack, the non-dimensionalised lift distribution will differ between model and original. Bayati et al. (2017) detail a scaling approach designed to ensure comparable non-dimensionalised blade loads. In this approach, the model chord distribution $c_M$ is calculated as

$$c_M = \frac{c_O}{\lambda_L} \frac{Kl_O}{Kl_M} \tag{1}$$

where $c_O$ is the original chord distribution, $\lambda_L$ is the geometric scaling factor, and $Kl_O$ and $Kl_M$ are the lift slopes in the
linear region of the original and model airfoil polars, respectively. The model twist distribution $\beta_M$ is calculated as

$$\beta_M = \beta_O - \frac{c_{l,O}^0}{Kl_O} + \frac{c_{l,M}^0}{Kl_M} \tag{2}$$

where $\beta_O$ is the original twist distribution and $c_{l,O}^0$ and $c_{l,M}^0$ are the lift coefficient values at zero angle of attack of the original and model airfoil polars, respectively.

Since the IEA 15 MW RWT's blade is very slender, applying this scaling approach leads to blades with very small chord
values. Using the $SD7032$ airfoil to create the geometry, such low chord values entail very thin blades. To avoid unnecessary challenges during the manufacturing process of the wind tunnel model blades, a constant factor is applied to the chord scaling so that

$$c_M = \frac{c_O}{\lambda_L} \frac{Kl_O}{Kl_M} C_c \tag{3}$$

with $C_c = 1.5$. Furthermore, this factor ensures angles of attack well away from the stall margin of the *SD7032* airfoil. Inboard
of $r/R = 0.25$, the chord is manually reduced to obtain a cylindrical root section with $D_{root} = 4\,\text{cm}$.

Rather than matching the lift force, a comparable thrust distribution along the blade is targeted. Therefore, equal thrust coefficient distributions $C_T = \frac{F_N\,dr}{\frac{1}{2}\rho U_\infty^2 2\pi r\,dr}$ are enforced

$$\frac{F_{N,M}}{\rho U_{\infty,M}^2 \pi r_M} = \frac{F_{N,O}}{\rho U_{\infty,O}^2 \pi r_O} \tag{4}$$

where $F_N$ is the axial force per unit span, $\rho$ is the density of air, $U_\infty$ is the freestream velocity and $r$ is the radial coordinate.
The local axial force can also be expressed as

$$F_N = L\cos(\phi) + D\sin(\phi) = \frac{1}{2}\rho V_{rel}^2 c\left(c_l\cos(\phi) + c_d\sin(\phi)\right) \tag{5}$$

with $\phi$ being the local inflow angle, $L$ and $D$ being the lift and drag force, and $c_l$ and $c_d$ the lift and drag coefficients, respectively. Substituting Equation 4 in Equation 5 yields a minimum function with $\beta_M$ as variable

$$\min_{\beta_M} = c_{l,M}(\phi_O - \beta_M)\cos(\phi_O) + c_{d,M}(\phi_O - \beta_M)\sin(\phi_O) - \frac{U_{\infty,M}^2}{U_{\infty,O}^2} \frac{r_M}{r_O} \frac{F_{N,O}}{\frac{1}{2}\rho V_{rel,M}^2 c_M} \tag{6}$$





where $V_{rel,M} = \sqrt{(U_{\infty,M}(1 - a_O))^2 + (\omega_M r_M (1 + a'_O))^2}$, with $a_O$ and $a'_O$ being the axial and tangential induction factors, respectively, and $\omega$ the angular rotation frequency. Based on Equation 6, the model twist distribution can be determined. The original flow properties $\phi_O$, $F_{N,O}$, $a_O$ and $a'_O$ are taken from numerical simulations of the full-scale IEA 15 MW RWT based on blade element momentum theory (BEM). For these simulations, an inflow velocity of $U_{\infty,O} = 10$ m/s is chosen, corresponding to operation just below rated. $U_{\infty,M}$ is set to match the targeted wind tunnel inflow velocity. The resulting chord

and twist distribution of the wind tunnel model blade are given in Figure 2.

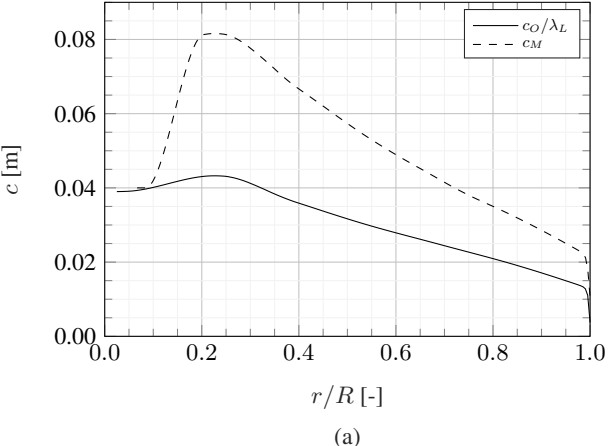
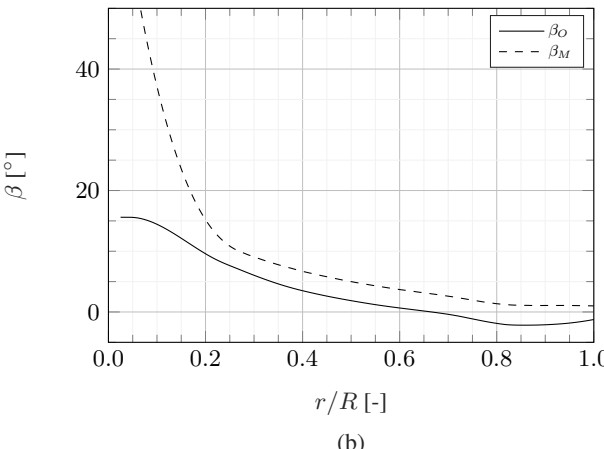

**Figure 2.** Chord (a) and twist (b) distribution of the geometrically scaled IEA 15 MW RWT and the wind tunnel model

### 2.2 Experimental setup and measurement system

The experiments were conducted in the Open Jet Facility at the TU Delft Faculty of Aerospace Engineering, which is a closed-circuit open jet wind tunnel. The jet exit is an octagon of $2.85\,\text{m} \times 2.85\,\text{m}$. A schematic of the OJF is given in Figure 3. The turbine was operated at an approximate tip speed ratio of $\lambda = 9$ and an inflow velocity of $U_\infty = 3.75$ m/s. Inflow conditions of

the wind tunnel were logged for each measurement point and showed no significant variation. The wind tunnel was kept at a constant temperature of $20\,^\circ\text{C}$.

In this campaign, SPIV was used to non-intrusively measure the flow around the blades. A Quantum Evergreen double-pulsed Neodymium-doped Yttrium Aluminium Garnet (Nd:YAG) laser provides the light source. Using laser optics, a thin vertical laser sheet was generated that illuminates the area around the targeted blade cross-section. To reduce reflections of the

laser, the blades and most other turbine components were spray-painted matt black. A Safex smoke generator produced smoke particles with a median diameter of $1\,\mu\text{m}$, which were used as tracers. The smoke generator was placed downstream of the tunnel test section, ensuring homogeneous mixing during the flow recirculation.

Two LaVision Imager sCMOS cameras with lenses of 105 mm focal length and an aperture of $f/8$, captured the illuminated particles during the two laser pulses. The laser and cameras were simultaneously triggered by an optical sensor that was



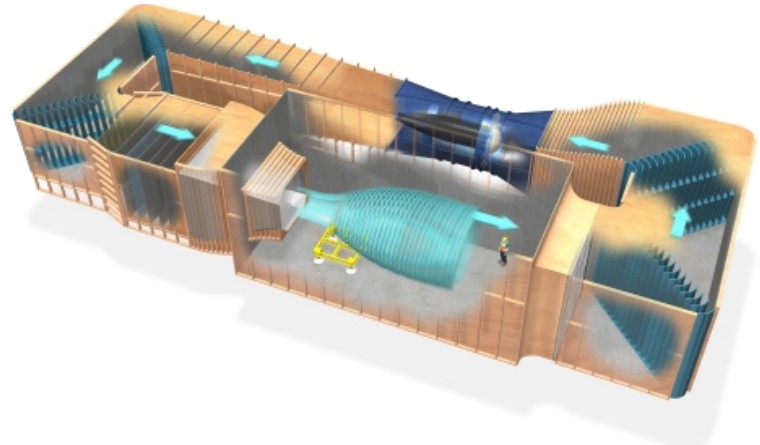

**Figure 3.** Schematic of the Open Jet Facility at TU Delft

activated by a notch in the rotor shaft once per revolution. A time delay between the optical sensor's signal and the laser/camera
trigger ensured the blade was in the horizontal position during its upward movement when taking the images. The image pairs
were taken with a time separation of $150\,\mu s$, which allowed the tracing of the particles' movement. This time separation is
equivalent to a particle movement of approximately $5\,px$ and a turbine rotation of $0.3°$. At each measurement location, 120
phase-locked images were taken, which are used in postprocessing to obtain an average flow field and its standard deviation.
The images are acquired and processed using the LaVision Davis 8 software. The field of view (FOV) resulting from this
measurement setup is approximately $FOV \approx 297\,mm \times 257\,mm$ and the final image resolution is $8.81\,px/mm$.

Both cameras and laser were mounted rigidly on a traversing system. This way, velocity measurements could be conducted
at multiple radial stations without the need for refocusing the cameras and calibrating the software. Figure 4 shows a schematic
of the measurement setup.

A total of 22 measurement planes were placed along the blade span as follows:

– $\Delta r/R = 0.100$ for $0.10 \leq r/R \leq 0.40$

– $\Delta r/R = 0.050$ for $0.40 \leq r/R \leq 0.80$

– $\Delta r/R = 0.025$ for $0.80 \leq r/R \leq 1.05$

This selection aims at accurately representing the stronger gradients in blade aerodynamics typically present close to the tip.
At four radial locations, namely at $r/R = [0.4,\ 0.6,\ 0.8,\ 0.9]$, measurements were taken for all three blades to evaluate how
representative the main measurement blade is for the remaining two blades.

When illuminating a cross-section, the blade cast a shadow where no particles could be traced. Thus, the flow field was
captured in two steps. In a first step, the blade's pressure side was evaluated by placing the laser upstream of the turbine and
angling the laser sheet downstream. Following that, the laser was relocated downstream of the rotor plane and its laser sheet



150 was tilted upstream to capture the suction side (as shown in Figure 4). In a postprocessing step, the two individual images were stitched together, resulting in the entire flow field around a blade cross-section.

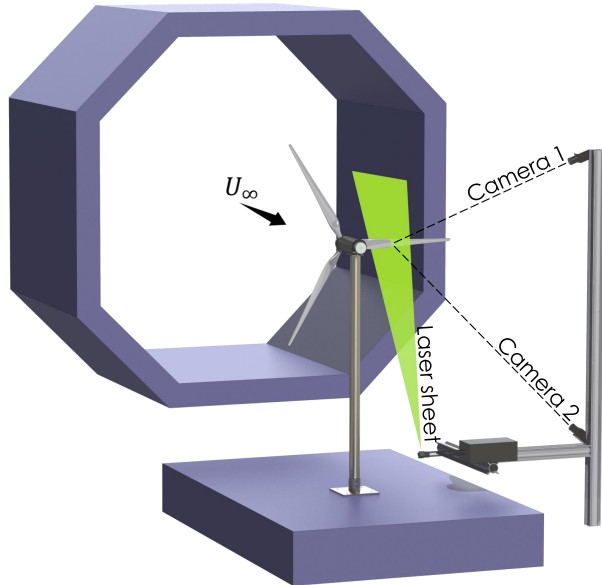

**Figure 4.** Experimental setup and measurement system

## 2.3 Deriving blade level aerodynamics from PIV measurements

This section presents the equations used to derive the distributed blade aerodynamics regarding bound circulation, induction, inflow angle and angle of attack, and blade loads. Based on these quantities, it is possible to calculate the experimental lift

155 polar, too. In this study, the equations presented below are applied under the assumption of local two-dimensional flow, i.e. only the velocity components in the measurement plane are considered.

### 2.3.1 Determination of bound circulation

The bound circulation $\Gamma$ at each measurement location can be calculated as the line integral of the measured velocity field $\boldsymbol{u}$ along a curve $S$ enclosing the blade cross-section (e.g. Anderson, 2017, p. 176):

$$
160 \quad \Gamma = -\oint_S \boldsymbol{u} \cdot \mathrm{d}s \tag{7}
$$

A study of the sensitivity to the bounding curve's size is presented in Appendix A. It revealed that the circulation, and also the forces calculated using Noca's method (see Section 2.3.3), do not exhibit perfect convergence with varying control volume size. As a consequence, the methods presented in this section are applied for multiple control volumes with different sizes, from which a mean value and standard deviation are calculated.





### 2.3.2 Determination of induced velocities, inflow angle and angle of attack

Several methods for determining the local inflow conditions exist. The inverse BEM approach (Bruining et al., 1993; Snel et al., 1994; Bak et al., 2006) uses measured/simulated forces as input to the blade element momentum equations and iteratively solves for the inflow conditions. Other methods characterise the inflow based on the annulus average flow field (Hansen and Johansen, 2004; Johansen and Sørensen, 2004) or based on the wake induction at the plane exactly between two blades (Herráez et al., 2018). Other approaches use the bound circulation strength to estimate local induced velocity and consequently the inflow conditions (Shen et al., 2007, 2009; Jost et al., 2018). Several benchmarks of these methods have been conducted based on CFD and/or experimental data (Guntur and Sørensen, 2014; Herráez et al., 2018; Rahimi et al., 2018).

The approach denoted as the Ferreira-Micallef method in Rahimi et al. (2018) is used here. It relies on potential flow theory to estimate the induced velocities at each spanwise location. This theory states that the velocity at any point can be expressed by the sum of the relative velocity and the velocities induced by free and bound vorticity, such that the measured velocity at a point $p$ is given as

$$\boldsymbol{u}_p = \sum \boldsymbol{u}_{ind} + \boldsymbol{V}_{rel} \tag{8}$$

Biot-Savart law is employed to determine the sum of the induced velocities at a set of control points located along $S$ so that

$$\sum \boldsymbol{u}_{ind} = \sum \frac{\Gamma}{2\pi} \frac{\boldsymbol{x}_p - \boldsymbol{x}}{|\boldsymbol{x}_p - \boldsymbol{x}|^2} \tag{9}$$

where $\boldsymbol{x}_p$ and $\boldsymbol{x}$ are the position vectors of the control point and inducing vortex element, respectively. By minimising the error between $\boldsymbol{u}_p$ and $\boldsymbol{u}_{ind}$ using a least squares approach, the relative inflow vector $\boldsymbol{V}_{rel}$ is determined, yielding the local axial and tangential induction factors

$$a = 1 - \frac{u_{rel}}{U_\infty} \tag{10}$$

$$a' = \frac{v_{rel}}{\omega r} \tag{11}$$

Knowing the induced velocities, the local inflow angle and angle of attack can then be calculated as

$$\phi = \tan^{-1}\left( \frac{U_\infty(1-a)}{\omega r(1+a')} \right) \tag{12}$$

$$\alpha = \phi - \beta \tag{13}$$

### 2.3.3 Determination of blade loads

*Noca's method*:

The forces exerted by an immersed body on the surrounding fluid can be evaluated by integrating the change of momentum over a finite control volume. Noca et al. (1999) presented an alternative formulation of the momentum conservation equation, solely relying on surface integrals of flow quantities placed on the boundary of the control volume. The forces can, thus, be





derived from the measured velocity field and its spatial and time derivatives. This approach has been successfully applied to PIV data collected on a vertical axis wind turbine by LeBlanc and Ferreira (2022). The force per density is given by:

$$\frac{\boldsymbol{F}}{\rho} = \oint_S \boldsymbol{n}\,\gamma\,\mathrm{d}s - \oint_{S_B} \boldsymbol{n}\,(\boldsymbol{u}-\boldsymbol{u}_B)^{\mathrm{T}}\,\boldsymbol{u}\,\mathrm{d}s - \frac{\mathrm{d}}{\mathrm{d}t}\oint_{S_B}\boldsymbol{n}\left(\boldsymbol{u}^{\mathrm{T}}\boldsymbol{x}\right)\mathrm{d}s \tag{14}$$

where $\boldsymbol{n}$ is the normal vector of the bounding curves, $\gamma$ is the flux term, $S$ is the outer boundary curve of the control volume surrounding the immersed body, $S_B$ is the control volume's inner boundary curve prescribed by the immersed body's surface, and $\boldsymbol{u}_B$ is the velocity vector of the immersed body's surface.

The term $\oint_{S_B}\boldsymbol{n}\cdot(\boldsymbol{u}-\boldsymbol{u}_B)\,\boldsymbol{u}\,\mathrm{d}s$ is related to the flow through the inner boundary curve $S_B$. Given the solid airfoil surface, this term is zero. The third term $\frac{\mathrm{d}}{\mathrm{d}t}\oint_{S_B}\boldsymbol{n}\cdot(\boldsymbol{u}\boldsymbol{x})\,\mathrm{d}s$ describes the force due to acceleration of the inner boundary surface. As the model wind turbine was running at a constant speed during the experiment, the velocity of the airfoil representing the inner boundary surface can be approximated as constant within the measurement plane. Therefore, this term is zero, too. The aerodynamic force vector $\boldsymbol{F}$ non-dimensionalised by the fluid density $\rho$ can then be determined as

$$\frac{\boldsymbol{F}}{\rho} = \oint_S \boldsymbol{n}\,\gamma\,\mathrm{d}s = \frac{1}{2}\oint_S \boldsymbol{n}\,u^2\boldsymbol{I}\,\mathrm{d}s - \oint_S \boldsymbol{n}\,\boldsymbol{u}^{\mathrm{T}}\boldsymbol{u}\,\mathrm{d}s - \frac{1}{\mathscr{N}-1}\oint_S \boldsymbol{n}\,\boldsymbol{u}^{\mathrm{T}}\left(\boldsymbol{x}\times\boldsymbol{\omega}\right)\mathrm{d}s + \frac{1}{\mathscr{N}-1}\oint_S \boldsymbol{n}\,\boldsymbol{\omega}^{\mathrm{T}}\left(\boldsymbol{x}\times\boldsymbol{u}\right)\mathrm{d}s$$

$$- \frac{1}{\mathscr{N}-1}\oint_S \boldsymbol{n}\left(\boldsymbol{x}\cdot\frac{\partial\boldsymbol{u}}{\partial t}\right)^{\mathrm{T}}\boldsymbol{I}\,\mathrm{d}s + \frac{1}{\mathscr{N}-1}\oint_S \boldsymbol{n}\,\boldsymbol{x}^{\mathrm{T}}\frac{\partial\boldsymbol{u}}{\partial t}\,\mathrm{d}s - \oint_S \boldsymbol{n}\left(\frac{\partial\boldsymbol{u}}{\partial t}^{\mathrm{T}}\boldsymbol{x}\right)\mathrm{d}s$$

$$+ \frac{1}{\mathscr{N}-1}\oint_S \boldsymbol{n}\left[\boldsymbol{x}\cdot(\boldsymbol{\nabla}\boldsymbol{\tau})\right]\boldsymbol{I}\,\mathrm{d}s - \frac{1}{\mathscr{N}-1}\oint_S \boldsymbol{n}\,\boldsymbol{x}^{\mathrm{T}}\left(\boldsymbol{\nabla}\boldsymbol{\tau}\right)\mathrm{d}s + \oint_S \boldsymbol{n}\,\boldsymbol{\tau}\,\mathrm{d}s^* \tag{15}$$

where $\boldsymbol{I}$ is the identity matrix, $\mathscr{N}$ is the dimensional constant, $\boldsymbol{\omega}$ is the vorticity vector and $\boldsymbol{\tau}$ is the Reynolds stress tensor.

There exist two possible frames of reference in which to apply the equations given above. On the one hand, a stationary reference frame can be chosen, where the measured blade cross-section moves vertically through the control volume, see Figure 5 (a). On the other hand, a reference frame rotating with the investigated cross-section can be used, see Figure 5 (b). While the original PIV data is captured in a stationary reference frame, it can easily be converted to a rotating frame by adding the apparent rotational velocity $V_{rot} = -\omega\,r$ to the measured vertical velocity component $v$.

For the analysis performed in the present work, a rotating frame of reference is chosen. In this reference frame, the time derivatives of Equation 15 are zero.

*Kutta-Joukowski theorem (KJ)*:

Alternative to Noca's method, the forces can be derived from the bound circulation using the Kutta-Joukowski theorem (e.g. Anderson, 2017, p. 282), which states that the sectional lift force is given by $L = \rho\,V_{rel}\,\Gamma$. This formulation can be decomposed

---

*Equations 14 and 15 require various vector multiplications. By default, these vectors are row vectors $\boldsymbol{\phi} = [\phi_1\ \phi_2\ \phi_3]$. Consequently, transposed vectors $\boldsymbol{\phi}^{\mathrm{T}}$ are column vectors.





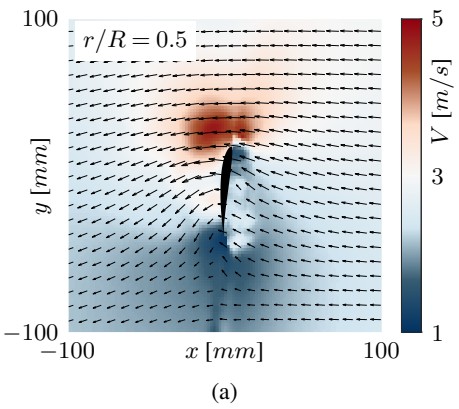
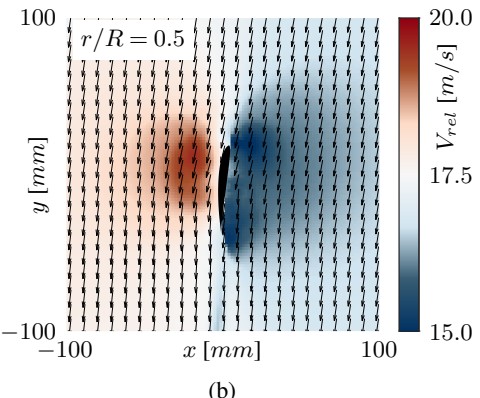

(a)
(b)

**Figure 5.** Velocity field in a stationary (a) and rotating (b) reference frame

to yield the forces normal and tangential to the rotor plane:

$$F_N = \rho \omega r (1 + a') \Gamma \tag{16}$$

$$F_T = \rho U_\infty (1 - a) \Gamma \tag{17}$$

It should be noted that the Kutta-Joukowski theorem is based on potential flow theory. Thus, e.g. the viscous drag contribution to the tangential force is neglected.

## 3    Results

### 3.1    Determination of the combined pitch and twist offset

The blades used in this experiment are made of vacuum-infused carbon fibre-reinforced material. This partially manual man-
ufacturing approach led to minor differences between the three blades. Based on visual inspection, one blade was chosen on which the measurement campaign was mainly conducted, hereafter called blade 1. However, measurements were taken for blades 2 and 3 at $r/R = [0.4, 0.6, 0.8, 0.9]$ to estimate the main measurement blade's representation of the other two blades.

Early investigations into the gathered data indicated non-negligible differences in blade aerodynamics between the three blades. To explain this behaviour, the blade cross-sections visible in the raw images were visually inspected and compared
against the original design of the blade. This approach is visualised in Figure 6 (a), where the blade cross-section is illuminated in white. The original design is overlaid as red airfoil shape. Then, the correct local twist is found by rotating this airfoil around the trailing edge until its pressure side follows approximately the same curve as the pressure side of the illuminated cross-section. This correction was determined with a precision of $0.1°$. The corrected airfoil is shown in green. Based on this comparison, it became apparent that the blade cross-sections were positioned at different angles than designed, resulting in the
offset in twist/pitch shown in Figure 6 (b).





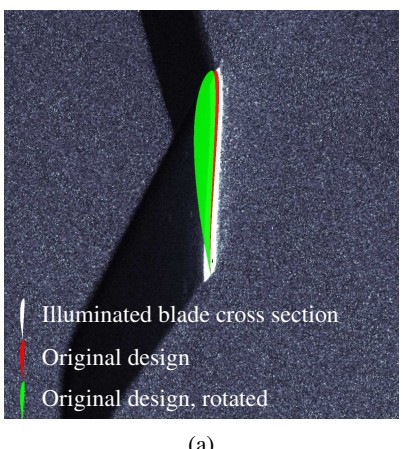
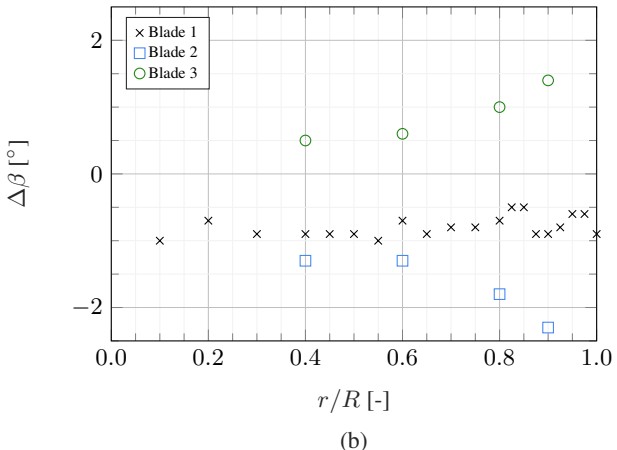

(a)              (b)

**Figure 6.** Approach of determining actual local airfoil orientation (a), twist/pitch offset determined by comparing experimentally captured blade cross-sections to the original design (b)

For blade 1, where many data points are available along the span, a quadratic fit is used to describe the trend and balance out the fluctuations likely due to human error in the interpretation of the raw images. Blade 1 appears to have a pitch offset of approximately negative one degree and additionally shows slight twist deformation towards the tip. More extreme twist deformations can be observed for blades 2 and 3, with opposite directions. This shows how challenging the use of vacuum-infused carbon fibre composite blades is. Despite having the same fibre layup, the manufacturing process is a highly manual task where minor differences can impact the structural properties of the blade. The pitch offset can be explained by the model turbine's connection between blade root and hub: The turbine is equipped with a manual pitch mechanism which is fixed in the desired position using set screws. Despite being used with care, this manual mechanism is likely the origin of the pitch deviations between the three blades.

## 3.2 Flow field

The flow fields represent the primary data collected during this experiment using stereoscopic PIV. Figure 7 depicts the measured velocity magnitude fields at the four radial stations where data for all three blades is available. Overall, the general flow patterns are in good agreement. However, the twist/pitch offset described in the previous section leads to differences in the angle of attack, explaining minor discrepancies in velocity magnitudes. For example, blade 2, exhibiting twist deformations towards higher angles of attack, induces higher velocities, while the opposite holds for blade 3.

Notably, many measurement points have low-velocity regions close to the suction side surface. Here, laser reflections from the blade surface reduce the accuracy of the PIV processing. This is less the case on the pressure side, where the concave blade surface causes lower reflections.







**Figure 7.** Non-dimensionalised velocity magnitudes at the radial stations measured for all three blades





### 3.3 Blade aerodynamics

All plots presented in this section contain error bars. These represent the 95% confidence interval and are based on variations in the measured velocity field during the capturing of the PIV images as well as in the processing with various control volume sizes, see Appendix A. This uncertainty is a measure of both the quality of the phase-lock as well as the unsteadiness of the flow. While almost all data points have very low uncertainty, the measurement point closest to the root suffers from the laser reflecting off the nacelle and hub, increasing measurement uncertainty. This effect is visible to a varying degree in all derived

aerodynamic quantities.

Figure 8 shows the circulation distribution of the three blades. The effect of varying pitch angles and twist deflection expresses itself in the different circulation levels of the three individual blades.

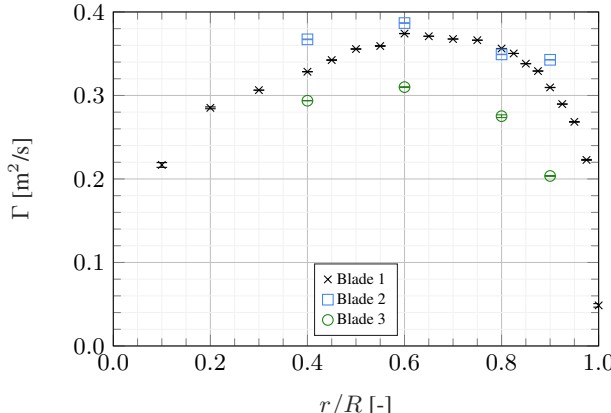

**Figure 8.** Spanwise distribution of bound circulation, error bars representing the 95% confidence interval

The axial and tangential induction factor distribution is shown in Figure 9. Compared to the circulation distribution, differences in induction are minor between the three blades. This is a significant finding in support of fundamental BEM theory,

which uses a rotor-averaged induction factor.

Figure 10 depicts the local inflow angle and angle of attack distribution. Given that the blades have a cylindrical cross-section at $r/R = 0.1$, the value of the angle of attack at this location is meaningless and reported for completeness only. The angle of attack distribution is evidently influenced by the pitch/twist variations between the three blades. Despite these variations, all derived angles of attack are well within the linear region of the design airfoil's lift polar.

The axial and tangential force distributions are presented in Figure 11. Two methods are employed to derive the normal force distribution, namely Noca's method and the Kutta-Joukowski theorem (KJ). Both methods are in close agreement; a linear fit between the results of all three blades yields $F_{N,KJ} = 1.016\,F_{N,Noca} - 0.3918$ with $R^2 = 0.9965$. By integrating the normal force distribution, the rotor thrust can be calculated and non-dimensionalised to obtain the thrust coefficient. To this end, piecewise cubic curves are fit to the experimental results. Where no data is available at blade root and tip, zero loading

is assumed. The resulting thrust coefficients are $C_{T,Noca} = 0.8170$ and $C_{T,KJ} = 0.7821$. For a tip speed ratio of $\lambda = 9$, the





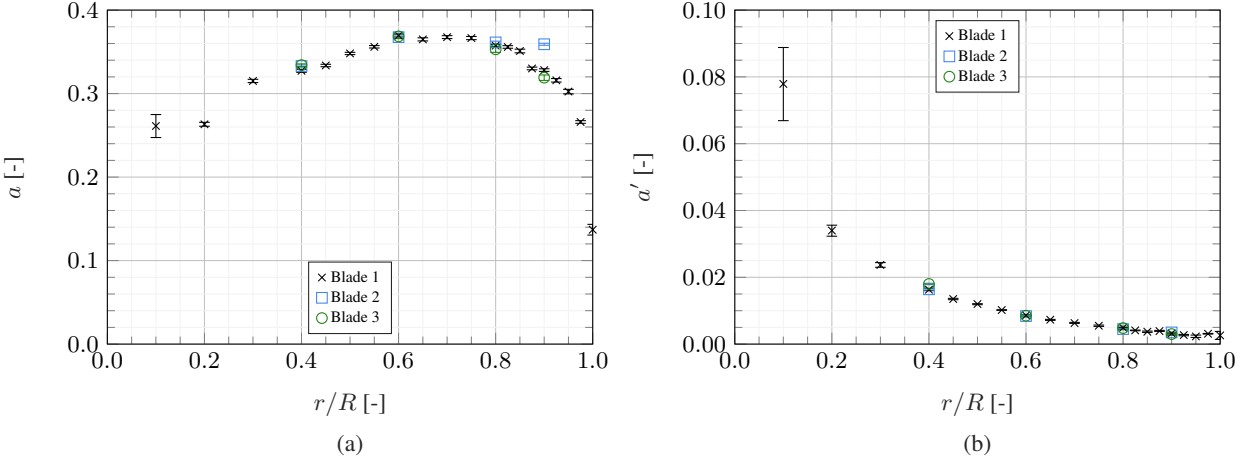

**Figure 9.** Spanwise distribution of axial (a) and tangential (b) induction factors, error bars representing the 95% confidence interval

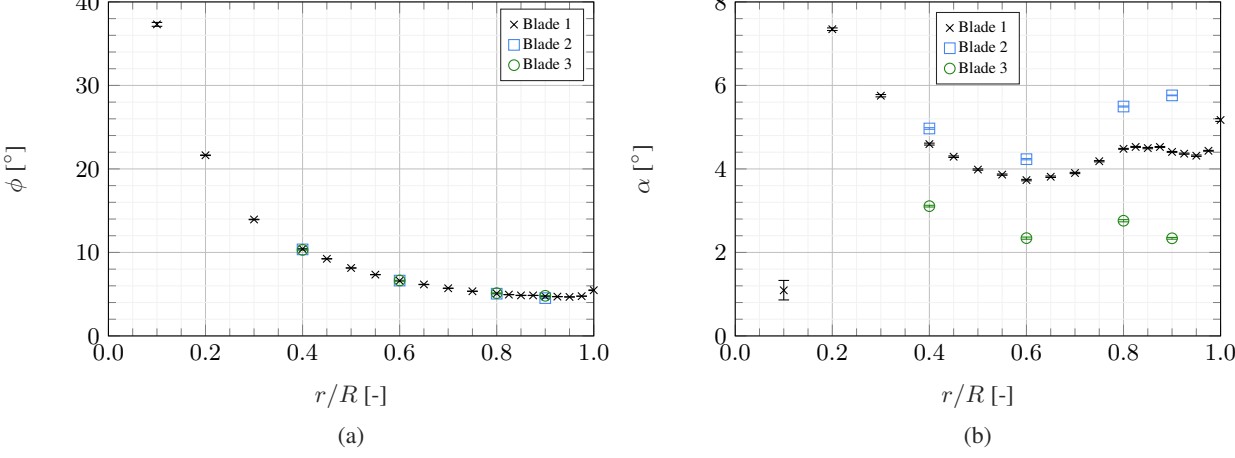

**Figure 10.** Spanwise distribution of inflow angle (a) and angle of attack (b), error bars representing the 95% confidence interval

IEA 15 MW RWT has a thrust coefficient of $C_T = 0.8$ (Gaertner et al., 2020). Thus, the relative deviation of the thrust-scaled blades to their reference corresponds to $\Delta C_{T,Noca} = 2.1\%$ and $\Delta C_{T,KJ} = -2.2\%$, respectively.

As demonstrated in Appendix A, Noca's method is, however, unreliable when estimating the tangential force from this experimental dataset. Therefore, only the tangential force derived using the Kutta-Joukowski theorem is presented here. It is
noteworthy, that this method neglects viscous effects and, consequently, misses the contribution of the viscous drag. Overall, the normal and tangential force trends are consistent between the three blades. However, the magnitude is fairly different, with blade 2 having, on average, slightly higher values than blade 1, while blade 3 exhibits lower values than the other two blades. These differences are in line with the pitch/twist offset discussed in Section 3.1.





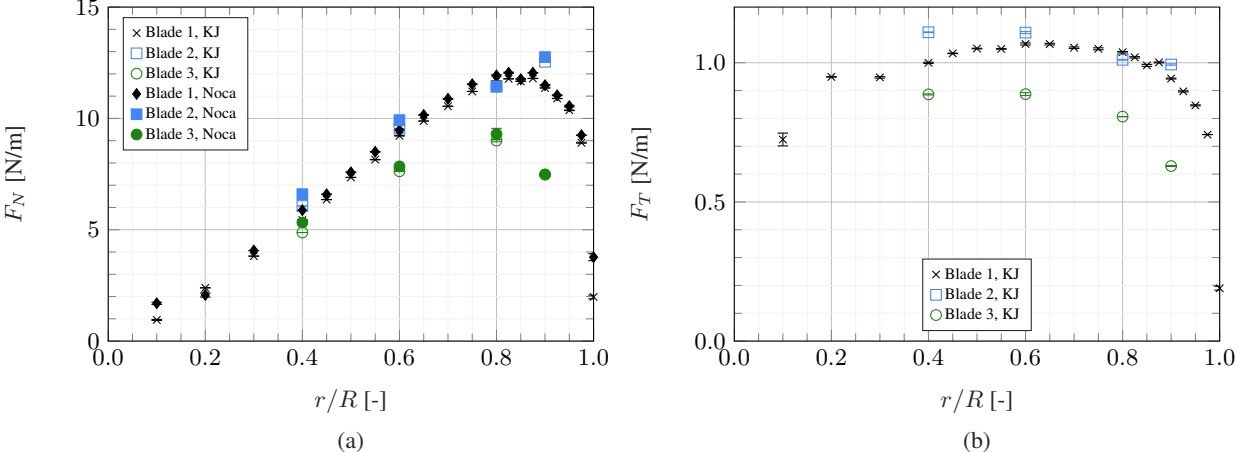

**Figure 11.** Spanwise distribution of normal (a) and tangential (b) force, error bars representing the 95% confidence interval

### 3.4 Lift polar

Based on the aerodynamic quantities presented in the previous section, the lift coefficient is derived. The lift force is calculated using the force distributions based on the Kutta-Joukowski theorem

$$c_l = \frac{F_{N,KJ}\cos(\phi) + F_{T,KJ}\sin(\phi)}{\frac{1}{2}\rho V_{rel}^2 c} \tag{18}$$

Figure 12 shows the experimental lift polar compared to the *SD7032* airfoil at $\mathrm{Re}_c = 50000$ (Fontanella et al., 2021b), resembling the Reynolds numbers present in this experiment, which varies between approximately 40000 and 65000 depending
on the radial position. For clarity, only the mean values are reported. The two measurements closest to the root are omitted as these cross-sections are defined by a cylinder and a blend between a cylinder and the *SD7032* airfoil. Additionally, the two measurements closest to the tip are omitted because the tip vortex causes highly three-dimensional flow features, which should not be compared to two-dimensional airfoil polars. The remaining measurement points are in good agreement with the lift coefficient curve of the design airfoil.

### 4 Conclusions and outlook

This study presents the results from an experimental campaign on a thrust-scaled version of the IEA 15 MW RWT. Particle image velocimetry is used to measure the flow field at multiple radial stations around the blade. Various aerodynamic blade properties are derived directly from the measured flow field along a closed curve around the blade cross-sections: The circulation is determined from the velocity integral, the inflow conditions by removing the blade induction from the measured flow
field using elemental potential flow solutions and the forces based on Noca's method and the Kutta-Joukowski theorem.





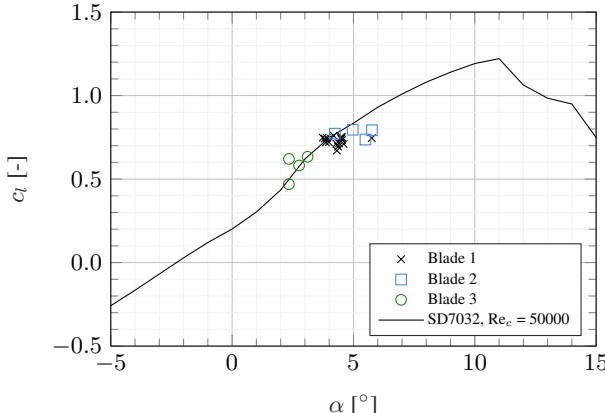

**Figure 12.** Experimental lift coefficient compared to the *SD7032* airfoil lift polar

Early analyses revealed that the blades were mounted with minor deviations from the desired pitch angle and, on top of that, exhibited twist deformations. This leads to considerable differences in the angle of attack and, consequently, blade loads among the three blades, which is consistently reflected in their experimentally derived spanwise distributions. In contrast, the derived induction values remain nearly constant between the three blades, indicating that induction can be considered a rotor-averaged

phenomenon. This is an experimental confirmation of one of the fundamental assumptions in blade element momentum theory.

The dataset created in this wind tunnel experiment fully characterises the three blades in terms of the surrounding flow field, bound circulation, local inflow conditions and blade loads. The normal force distributions derived using Noca's method and the Kutta-Joukowski theorem were found to be in good agreement. Knowing these aerodynamic parameters, it can be demonstrated that the lift coefficient measured along the span agrees well with the lift polar of the airfoil used in the blade design.

The experimental data presented here can be used in future numeric model validation studies. It provides data relevant for validating low-fidelity models, such as algorithms based on blade element momentum theory or lifting line theory, and for mid to high-fidelity models, such as panel codes and computational fluid dynamics. Since the model blade is based on the IEA 15 MW RWT, the non-dimensionalised loads resemble the current state-of-the-art of real offshore wind turbines and numerical reference models. Furthermore, the newly created model wind turbine can be used in future experiments investigating the

aerodynamics of this reference wind turbine.

To reduce the impact of blade deflections in future research, it is recommended to either produce a new set of blades with less variation in their stiffness properties or to apply more advanced deformation tracking techniques such as photogrammetry. To improve the accuracy in pitch setting, the manual pitch mechanism could be exchanged for a variable pitch mechanism controlled by a motor. This would then require an initial calibration before a new experimental campaign.





*Data availability.* The data presented in this study, as well as information regarding the blade planform and logged wind tunnel operating conditions, are openly available on the 4TU.ResearchData repository at DOI:10.4121/164890ab-39d7-4af8-8b3c-9e21f789b80a.

## Appendix A: Sensitivity to chosen control volume

In this study, the blade's aerodynamic quantities are determined by interrogating flow information along a closed curve enclosing the investigated blade cross-section. A circular curve is chosen with the blade cross-section positioned in its centre. To

verify the methods presented in Section 2.3, a panel code developed by Ribeiro et al. (2022) based on the work of Katz and Plotkin (2001) is used to replicate the wind tunnel experiment numerically. The panel code simulates the three-dimensional surface of the blade and can be used to derive flow fields at locations equivalent to the measurement planes of the experiment. Such results then offer the opportunity to derive circulation and loads based on the velocity field around the blade ("indirect") but also from the aerodynamic solution on the blade ("direct"). By comparing these two approaches, the methods for deriving

aerodynamic quantities from the flow field can be verified before applying them to the experimental data.

Figure A1 shows the sensitivity of the calculated circulation and of the forces based on Noca's method to the control volume's size, given as the ratio of its radius $r_{CV}$ to the local chord, at three radial locations. When calculating the forces based on the Kutta-Joukowski theorem, they are directly proportional to the circulation distribution and are, thus, not presented here.

The sensitivity is investigated for both the experimental data as well as the panel code results. Generally, there is a conflict

of interest between the data points per control volume size which favours a large control volume and the approximation of two-dimensional flow in a flat measurement surface which favours a small control volume.

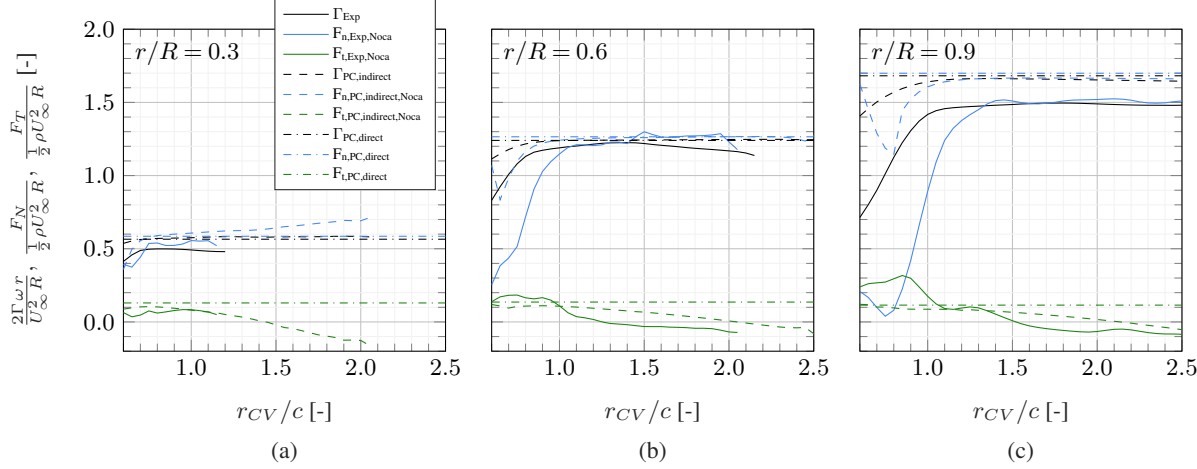

**Figure A1.** Sensitivity of determined blade loads and circulation to the chosen boundary curve size at various radial stations

For the panel code results (PC), it can be observed that the indirectly determined circulation converges against the directly determined value with increasing control volume size. For the normal force, this is only true for the two outboard sections



shown in Figure A1 (b) and (c). The discrepancy between the direct and indirect approach at the inboard section can be
attributed to the increasing flow curvature in this region, which stands in contrast to the two-dimensional control volume.

In contrast to circulation and normal force, the tangential force does not converge anywhere along the span but rather
decreases with increasing control volume size. The high tip speed ratio of the model turbine entails very low torque values and
the tangential force is very small. As such, the momentum change corresponding to the tangential force is difficult to capture
with the Noca method. Based on this finding, only the tangential force calculated via the Kutta-Joukowski theorem is presented
in this article.

The circulation and forces determined based on the experimental data largely follow the same trends observed for the panel
code results. However, given the less clean flow field, the convergence is not as steady and shows slight deviations even after
the initial, clearly unconverged, ramp. This is particularly true for the forces calculated using Noca's method, which relies on
sensitive derivatives of the velocity field. To limit the influence of the control volume, the convergence is evaluated individually
for each measurement plane and the endpoint of the initial convergence ramp is identified. The aerodynamic quantities are
determined for multiple control volumes with sizes beyond the initial convergence ramp and then averaged over these. This
approach yields the results presented in Section 3.3. It should further be noted, that for the experimental results, the largest
possible control volume is dictated by the available field of view. Thus, the convergence of methods such as Noca's should be
taken into consideration when defining the PIV setup and, consequently, the field of view.

*Author contributions.* EF designed the wind turbine model, built the model blades, planned and executed the experiment, and post-processed
and analysed the measurement data. AR contributed to the experiment execution and data analysis and provided numerical simulations to
help the development of post-processing methods. KB acquired funding and contributed to the experiment planning and the data analysis.
CF acquired funding and contributed to the experiment planning and execution, the development of post-processing methods, and the data
analysis.

*Competing interests.* The authors declare that they have no competing interests.

*Acknowledgements.* This contribution has been financed with Topsector Energiesubsidie from the Dutch Ministry of Economic Affairs under
grant no. TEHE119018. The wind tunnel experiment was financed by internal TU Delft funding.





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
