# Peer review of "Aerodynamic characterisation of a thrust-scaled IEA 15 MW wind turbine model: Experimental insights using PIV data"

_Wind Energy Science, 2024_

## Author Comment (AC1)

Dear reviewers,

The authors would like to thank you for the time and effort that you have dedicated to providing valuable feedback on our manuscript. We have been able to incorporate changes to reflect your suggestions. Please find a point-by-point response to your comments and a version of our manuscript highlighting all made changes below.

**Reviewer 1**

1. Refer to the Table 2: The scaling factor for the rotor diameter D is 133.33, while for the root radius it stands at 50. Drawing from my experience, I speculate that the difference in the scaling factor could stem from the necessity to accommodate electronics within the rotor root. This adjustment may have disrupted the maintenance of a consistent scaling factor. However, such intricacies might not be readily apparent to individuals not directly involved in the experimental process. Hence, I advise to present the rationale behind the difference in the scaling factor. Furthermore, it is important to address the potential ramifications of this difference on the ultimate conclusions drawn from the study. To bolster these assertions, referencing relevant studies would provide additional support to your arguments.
   Your speculations are correct. This region requires mechanical parts strong enough to carry centrifugal loads and any imbalances in the rotor. Therefore, the mechanical components cannot be scaled by the same factor as the rotor itself. A study of existing literature reveals that the root radius is almost never reported and/or the chord distribution is plotted all the way to $r/R = 0$, which is unrealistic. One example reporting the root radius is the work by Fontanella et al. (2022). Here, a ratio of root to tip radius of 0.075 is reported, while ours is 0.067. We have referenced their work, see Page 5, Line 244. Regarding the impact on the studies result, see our answer to your next comment, which we feel is closely related.

2. Refer to the line 100: In the study, the authors noted a manual reduction of the chord to 4cm below r/R = 0.25. However, the term "manually" lacks clarity regarding its intended meaning. It appears to indicate a deviation from utilizing the scaling law specified in equation 3. If this interpretation holds true, what alternative method did the authors employ? Furthermore, what factors contribute to their confidence that the results would not significantly diverge from those obtained by precisely following equation 3? The authors are encouraged to expound upon their approach, providing support from relevant literature.
   We agree that the term manually is not precise enough here. We used a cubic spline to reduce the chord to the cylindrical root region, as explained now on Page 6, Line 273. This reduction of the chord is common procedure for wind tunnel experiments. While not always explicitly reported, it can be deducted from images/graphs that this is done for most of the papers referenced in our literature review. It is expected to have little impact on rotor aerodynamics since the aerodynamic forces in this region are low when compared to the outboard region. This line of argumentation has been added to the text as well.

3. Upon re-deriving equation 6, I discovered that it was not that straightforward. The derivation of this equation entails several algebraic steps. Given the significance of this equation in determining the blade's twist angle, it would be beneficial to include its derivation in either the main text or an appendix. Additionally, I recommend using the term "twist angle" to define $\beta$, as "pitch angle" typically denotes the rotation of the entire blade at the root along the blade axis with respect to the rotor plane.
   Thank you for this comment. We can see that the derivation of the equation should be made more clear to the reader. Therefore, we have added additional algebraic steps between Page 6, Line 278 and Page 7, Line 292. Furthermore, we have changed the definition of $\beta$ in the nomenclature to twist angle.

4. Refer to the line 112: The authors are requested to precisely indicate the simulations from which the parameter values were extracted. Furthermore, it is advisable to provide insight into the reasons behind selecting those specific papers and elaborate on their relevance to the current study.
   The simulations were conducted using an inhouse BEM algorithm developed by the authors. The algorithm has been

used in published work to simulate the IEA 15 MW RWT with straight and swept blades. In that work, the straight blade simulations were validated against the established TNO-inhouse lifting line algorithm AWSM. Thus, we have confidence in the accuracy of the numerical results used in the scaling approach presented in this study. We have specified the used algorithm in the text, see Page 7, Line 296.

5. Refer to lines 130 and 131: Please specify the value of the time delay and the way the authors calculated it.

The time delay value was 41 ms. It was determined (rather than calculated) by comparing images captured during turbine operation to standstill images with the blade fixed in the desired position. By conducting this comparison close to the tip, where the sensitivity to the time delay is largest due to the high rotational velocity, we ensured that all measured cross-sections were in the desired rotor position. We have added this information to the text, see Page 8, Line 320.

6. Suggestion for Equation 14: It is noted that you have presented the equation in vector form. For mathematical accuracy, it would be more appropriate to use $n \cdot (u - u_B)$ instead of $n(u - u_B)^T$. This adjustment should also be applied to the third term. While Euclidean inner products, which are relevant in this context, can be calculated as the matrix product of a row and column vector, they are distinct concepts in precise mathematical terms.

Thank you for pointing this out. We have applied the suggested changes to Equation 14. We have additionally adjusted Equation 15 to be aligned with the original paper by Noca.

7. Refer to the Figure 12:

   (a) Blade 3: If the airfoil maintains a consistent shape across all sections, one would expect identical lift coefficients (cl) for a given Angle of Attack (AOA). However, a difference is observed in the initial two points, suggesting other factors at play. It would be beneficial to explore potential causes for these differences. On a positive note, it is commendable that the remainder aligns well with the design lift curve.

   (b) Blade 2: The observation that the lift coefficient (cl) remains constant despite increasing AOA, especially at lower angles, is unexpected. This outcome warrants further investigation to determine the underlying cause, as it does not closely adhere to the anticipated design lift curve, leading to some reservations about its accuracy.

   (c) Blade 1: This blade shares the same issue as Blade 2, with an added discrepancy at a 6° AOA, where the cl value significantly deviates from the design lift curve. For Blades 1 and 2, it is plausible that the airfoil's shape diverges from the standard SD7032 profile, possibly in the curvature at the leading edge. Such variations might also stem from inadequate resolution in the velocity field analysis.

Thank you for this comment. We agree with your observations that some experimentally determined $c_l$ values deviate from the design polar or from their expected relative position to other measurement points. We believe there are two major factors influencing these results.

Firstly, the Reynolds number varies along the span such that a comparison between the experimental data and a design airfoil polar at a single Reynold number can only be drawn based on trends. This was also pointed out by your fellow reviewer, so please also refer to Reviewer 2, Comment 6 and our corresponding answer. The additional analysis proposed by Reviewer 2 led us to the new insight that there is a slight deficit in lift production between the blades used in this experiment and the design airfoil's polars in the root and tip region.

Secondly and as you mentioned already yourself, some of these discrepancies are likely due to small deviations in geometry that might have been introduced during the manufacturing process. Furthermore, the airfoil model on which the $SD7032$ polars were measured had a chord of 130 mm while the majority of our measurements are taken at sections with chord lengths smaller than 50 mm. With such dimensions, small errors in geometry have a relatively larger impact on the results. Lastly, potential differences in surface finish could also play a role. Based on these arguments, we believe that our analysis, while acknowledging existing deviations, still demonstrates a high level of accuracy in a challenging experimental setup.

8. Refer to lines 314 and 315: This scaled-down version of the original wind turbine (WT) replicates the non-dimensional thrust by design, as it was developed based on thrust similarity principles. However, the scaled-down WT is tailored to

a specific thrust configuration, determined by factors such as blade pitch angle (where pitch angle refers to the angle by which the entire blade rotates around its axis relative to the rotor plane). Altering the pitch angle of blades in the original WT would necessitate a redesign of the scaled WT. For instance, consider the minimum function defining the twist angle $\beta$ (as shown in equation 6). Changing the inflow angle to the blade through pitch adjustment would consequently alter the radial twist distribution in the scaled-down WT. It's important to exercise caution, as blade shapes and types differ between the original and scaled-down WTs, resulting in varying Reynolds numbers. Consequently, flow physics, such as flow transition and separation at a given radial position (r/R), differ at the airfoil level. Therefore, while global aerodynamic properties like non-dimensional thrust may exhibit similarity, it's essential to recognize that local aerodynamics at each radial position between the original and scaled-down WTs may not necessarily align.

We fully agree with your argumentation. We felt a statement along these lines was best placed at Page 8, Line 301 rather than in the conclusions section and hope you agree with this assessment.

**Reviewer 2**

1. In the introduction the authors mention other scaled experiments with e.g. the DTU 10 MW model turbine. There are also a few experiments with a model turbine scaled down from a 5MW reference turbine (https://doi.org/10.5194/wes-6-1341-2021). In these experiments they also use the method by Herreaz to determine the axial and tangential induction factor along the rotor blade.

   Thank you for pointing out this gap in our literature review. The research done at ForWind is relevant to this study and should be mentioned in the introduction. We have added references to relevant publications, see Page 4, Line 209. Since you mention their use of the Herraez method, let me explain two reasons why we did not choose this method. Firstly, we were interested in deriving blade loads from the flow field around the blade. Thus, we would have had to take twice as many measurements (or introduce a second measurement technique like LDA) if we also wanted to capture the flow in the bisectrix. Secondly, we ran an analysis very similar to the one presented here on a turbine with swept blades (currently under review: https://doi.org/10.5194/wes-2024-11). For swept blades, the symmetry condition of the Herraez method is not given anymore, and the method would fail. To ensure comparability between the two campaigns, we went with a method that was applicable to both.

2. Figure 3 can easily be removed since it doesn't provide any extra information for the rest of the paper.

   You are right. Upon reviewing this section, we agree that the figure indeed adds little content. We have removed the figure and the reference to it, see Page 8, Line 306.

3. On page 9 line 150 the authors state that in the post-processing they stich together the results from the pressure and the suction side of the profile to get the total flow around the profile. Are these images averaged over several events or are they temporally highly resolved? In the later case, can the authors say anything about variations in the flow field on the upper and lower side for measurements between single rotations?

   The former is the case. We first individually average the pressure and suction side flow fields over the 120 phase-locked images taken at these locations. Then, the two average flow fields are stitched. We have adjusted the text to be more precise about this procedure, see Page 9, Line 342.

4. Line 200, page 11: The model turbine was running at constant rotational speed. Was that actively controlled or is that given due to the fact that the inflow was constant ? I can image, that there might have been some fluctuations in the rotational speed also due to the fact that the blades were all different. Passing the tower will cause different aerodynamics for each of the blades which could be seen in fluctuations in the rotational speed. Did the authors observe something like that ?

   Indeed, we actively control the rotor speed. Instead of a pitching mechanism to regulate torque and rpm, the turbine is driven by a motor that closely follows the set rotational speed. In the future, we hope to expand the turbine's capability to be driven by its aerodynamic torque rather than a motor. We have added this information, see Page 8, Line 307. Given this setup, we did not observe azimuth-dependent variations in the rotor speed.

5. In section 3.1 it is not quite clear if the authors corrected the mean offset on the pitch (twist) for each blade or if they just measured the offset. If they did not correct the mean offset, why not? The authors say that the turbine is equipped with a manual pitch system which should allow the correction of the offset, right?

Thanks for pointing out that this is not fully clear. The differences in pitch angle and twist deformation were unfortunately only found in postprocessing after the campaign had ended. Thus, the pitch angle could not be corrected anymore. We have added a statement to clarify this, see Page 13, Line 439.

6. I do not fully understand why the authors show figure 12. Here they compare the calculated lift coefficient for each blade and compare it to the one from a reference for a specific Reynolds number. Why not going the other way and calculating the normal and tangential forces for each radial position on the blades using the angle of attack, acting velocities and the corresponding lift and drag coefficients from figure 1 for the correct Reynolds number and ad the results to figure 11? That would minimise the effect of the Reynolds number in the end should show how well all methods agree — or maybe I misunderstood the intension of figure 12.

The intention of this plot is to demonstrate that the experimentally derived lift polar follows the trend of the design airfoil's lift polar. We do, however, agree with you pointing out the dependency on the local Reynolds number. We would like to keep Section 3.3 unchanged as we would rather not mix up quantities directly derived from measurements with those indirectly derived (via the input polars). Additionally, Figure 10 (a) (formerly Figure 11 (a)) already contains six datasets and we do not want to overload this figure. Thus, we alternatively present a comparison between the $c_l$ derived from the measurements and that derived via the polars in Section 3.4, see Page 17, Line 491 and Figure 11 (b). This analysis reveals that there are small deficits in lift production in the root and tip region between the blades used in the experiment and what the polars would suggest. This is a new insight, and we would like to thank you for the suggestion that led to it. We have also added a statement in the conclusions section to reflect this insight, see Page 19, Line 513. To reflect the Reynolds number variation along the span, we have also included the SD7032 lift polars for two more Re numbers in Figure 11 (a). We hope these changes are in agreement with your suggestion.

7. Line 316, page 18, the authors mention that in future research they would like to reduce the impact of blade deflection on the results. In the paper they never mentioned the problem or determined the deflection and the impact on the results. While you mentioned it here, what is the impact of blade deflection on the results presented in the paper?

Firstly, we need to clarify that we meant deformations, but wrote deflections. To be more consistent with the rest of this paper, we have adjusted the text, see Page 19, Line 521. Secondly, we acknowledge that you raise a good point by asking about deflections. Maximum flapwise tip deflections were determined to be 14 mm. In relation to the 900 mm blade tip radius, the influence of these deflections on the aerodynamic analysis presented in this paper is considered negligible.

We would like to thank the reviewers again for their detailed and constructive feedback. Please find a version of our manuscript highlighting all the changes made on the following pages. We look forward to hearing from you in due time regarding our submission and to responding to any further questions and comments you may have.

Sincerely,
Erik Fritz, André Ribeiro, Koen Boorsma, Carlos Ferreira

[revised manuscript text omitted]
{1}{\mathcal{N}-1}\boldsymbol{u}\left(\boldsymbol{x}\times\boldsymbol{\omega}\right) + \frac{1}{\mathcal{N}-1}\boldsymbol{\omega}\left(\boldsymbol{x}\times\boldsymbol{u}\right) \\ & - \frac{1}{\mathcal{N}-1}\left(\boldsymbol{x}\cdot\frac{\partial\boldsymbol{u}}{\partial t}\right)\boldsymbol{I} + \frac{1}{\mathcal{N}-1}\boldsymbol{x}\frac{\partial\boldsymbol{u}}{\partial t} - \frac{\partial\boldsymbol{u}}{\partial t}\boldsymbol{x} \\ & + \frac{1}{\mathcal{N}-1}\left[\boldsymbol{x}\cdot(\boldsymbol{\nabla}\cdot\boldsymbol{\tau})\right]\boldsymbol{I} - \frac{1}{\mathcal{N}-1}\boldsymbol{x}\left(\boldsymbol{\nabla}\cdot\boldsymbol{\tau}\right) + \boldsymbol{\tau} \end{aligned} \tag{15}$$

400 EF

[revised manuscript text omitted]